# Testicular Localization and Potential Function of Vimentin Positive Cells during Spermatogonial Differentiation Stages

**DOI:** 10.3390/ani12030268

**Published:** 2022-01-22

**Authors:** Amirreza Niazi Tabar, Hossein Azizi, Danial Hashemi Karoii, Thomas Skutella

**Affiliations:** 1Faculty of Biotechnology, Amol University of Special Modern Technologies, Amol 4616849767, Iran; A.niazitabar@ausmt.ac.ir (A.N.T.); D.hashemi@ausmt.ac.ir (D.H.K.); 2Institute for Anatomy and Cell Biology, Medical Faculty, University of Heidelberg, Im Neuenheimer Feld 307, 69120 Heidelberg, Germany; Thomas.Skutella@uni-heidelberg.de

**Keywords:** vimentin, spermatogonia, stem cell, germ cells, male fertility

## Abstract

**Simple Summary:**

Spermatogonia stem cells (SSCs) in the testis are responsible for transmitting genetic information to subsequent generations. During the differentiation of SSCs, various cytoskeletons are involved in giving the cell shape and internal organization. One of the essential cytoskeletons that play functions in these spermatogenic processes is vimentin. This study examined the vimentin expression in vivo and in vitro by immunocytochemistry (ICC), immunohistochemistry (IMH), Fluidigm real-time polymerase chain reaction, and bioinformatics analysis. IMH analysis demonstrated that the high vimentin expression was localized in the middle and central testicular cords cells of seminiferous tubules and low expression in the basal region under in vivo conditions. To evaluate the expression of vimentin in vitro, we first isolated SSCs and then cultured cells from the testis. Our results showed that vimentin plays important roles in the differentiation of testicular germ cells.

**Abstract:**

Vimentin is a type of intermediate filament (IF) and one of the first filaments expressed in spermatogenesis. Vimentin plays numerous roles, consisting of the determination of cell shape, differentiation, cell motility, the maintenance of cell junctions, intracellular trafficking, and assisting in keeping normal differentiating germ cell morphology. This study investigated the vimentin expression in two populations of undifferentiated and differentiated spermatogonia. We examined vimentin expression in vivo and in vitro by immunocytochemistry (ICC), immunohistochemistry (IMH), and Fluidigm real-time polymerase chain reaction. IMH data showed that the high vimentin expression was localized in the middle of seminiferous tubules, and low expression was in the basal membrane. ICC analysis of the colonies by isolated differentiated spermatogonia indicated the positive expression for the vimentin antibody, but vimentin’s expression level in the undifferentiated population was negative under in vitro conditions. Fluidigm real-time PCR analysis showed significant vimentin expression in differentiated spermatogonia compared to undifferentiated spermatogonia (*p* < 0.05). Our results showed that vimentin is upregulated in the differentiation stages of spermatogenesis, proving that vimentin is an intermediate filament with crucial roles in the differentiation stages of testicular germ cells. These results support the advanced investigations of the spermatogenic process, both in vitro and in vivo.

## 1. Introduction

Spermatogenesis is a main genital process in the male reproductive system. The fundamental role of this system is performed by spermatogonia stem cells (SSCs) [1]. SSCs are located in the basement membrane of seminiferous tubules and supported by Sertoli cells, peritubular myoid cells, and other somatic cells. Sertoli cells (a kind of sustentacular cell) are part of a seminiferous tubule that supports spermatogenesis. These somatic cells support the fate of SSCs by extracellular signals and by secreting growth factors through spermatogenesis [2,3]. 

Spermatogenesis is a complex system that has the responsibility of transmitting genetic information to the next generation. In this process, spermatogonia differentiate and proliferate into spermatozoa. According to the topographical organization of undifferentiated spermatogonia, they are subdivided into single A (As), paired A (Apr), or aligned A (Aal). As to Aal spermatogonia are collectively called undifferentiated spermatogonia. These cells proliferate all through the section of the seminiferous epithelium cycle and then become quiescent before most of them differentiate into A1 spermatogonia. After a critical subsequent division (A2-B), differentiated SSCs divide into primary and secondary spermatocytes. After some meiotic divisions, secondary spermatocytes become round spermatids, which develop into elongating spermatozoa. A1 to B spermatogonia are collectively called differentiated spermatogonia [4,5].

Spermatogonia show two different activities. First, self-renewal by mitotic divisions to maintain the primary pool of stem cells, and second, spermatogenesis, defined as the differentiation of undifferentiated spermatogonia into differentiated spermatogonia [6]. These events are also related to widespread adjustments in the size and shape of germ cell movement related to cytoskeletal activity. The cytoskeleton, which accommodates actin, microtubules, and intermediate filaments, characterizes those cells’ activities [5,7]. One of the critical cytoskeletons that play crucial functions in these spermatogenic processes is vimentin [8,9].

Vimentin is kind of intermediate filament and one of the first filaments expressed in spermatogenesis. Vimentin is an intermediate filament that extends from the nuclear periphery to the cell membrane, and is connected with the tubulin and actin cytoskeleton [10]. The cellular functions of vimentin contribute to cellular stiffness, actin position, cell migration, cell division, and organelle organization in different stages of spermatogenesis [10,11]. In addition, vimentin plays other vital roles, such as determining cell shape, cell differentiation, cell motility, maintaining cell junctions, assisting in keeping ordinary spermatogonia morphology, and playing a main role in anchoring differentiating germ cells to the seminiferous epithelium [12,13]. 

Vimentin may have a critical role in SSCs differentiation; however, it is unclear if the vimentin intermediate filament is necessary during differentiation in vitro, and some research was conducted on stage association in the rat seminiferous epithelium [13,14]. Lastly, there are a few studies on vimentin expression in male germ cells. In this experimental study, we investigated the expression of vimentin in vivo and in vitro in seminiferous tubules and germ cells.

## 2. Materials and Methods

### 2.1. Testicular Cell Isolation

In this experimental study, protocols for animal care and surgical intervention of adult mice were approved by Amol University of Special Modern Technologies, Amol, Iran. Male mice (7 weeks, C57BL/6 mice) were purchased from Pasteur Institute (Iran). Mice testes were collected in phosphate-buffered saline containing 2% BSA and 0.1% Triton X100. The testes’ seminiferous tubules were minced into small pieces in Dulbecco’s modified Eagle’s medium (DMEM, Invitrogen, Waltham, MA, USA). Testicular cells were collected by enzymatic digestion in a solution containing dispase (0.5 mg/mL) (Sigma Aldrich, St. Louis, MO, USA), collagenase IV (0.5 mg/mL) (Sigma Aldrich, St. Louis, MO, USA) and DNase (0.5 mg/mL) (Sigma Aldrich, St. Louis, MO, USA) in an HBSS buffer with Ca^2+^ and Mg^2+^ (PAA, Whatman, MA, USA) at 37 °C for 10 min. Then, isolated testicular cells were washed with DMEM/F12 (Invitrogen, Waltham, MA, USA), and the cell suspension was passed through a 70 μm mesh filter and centrifuged at 1500 rpm in 10 min. The isolated testicular cells were kept at 37 °C in 5% CO_2_ in air. The culture medium was altered every third day [3,15].

### 2.2. Testicular Cell Culture

The adult testicular cell suspension was plated into 0.2% gelatin-coated plates in an SSC culture medium, which contained StemPro-34 medium, 5 µg/mL bovine serum albumin (Sigma Aldrich, St. Louis, MO, USA), 1% L-glutamine (PAA, Whatman, MA, USA), 0,1% ß-mercaptoethanol (Invitrogen, Waltham, MA, USA), 6 mg/mL D+ glucose (Sigma Aldrich, St. Louis, MO, USA), 1% nonessential amino acids (PAA, Whatman, MA, USA), 1% N2-supplement (Invitrogen, Waltham, MA, USA), 1% penicillin/streptomycin (PAA, USA), 1% MEM vitamins (PAA, USA), 60 ng/mL progesterone (Sigma Aldrich, St. Louis, MO, USA), 10 ng/mL FGF (Sigma Aldrich, St. Louis, MO, USA), 20 ng/mL epidermal growth factor (EGF), 100 µg/mL ascorbic acid (Sigma Aldrich, St. Louis, MO, USA), 30 ng/mL estradiol (Sigma Aldrich, St. Louis, MO, USA), 30 µg/mL pyruvic acid (Sigma Aldrich, St. Louis, MO, USA), 8 ng/mL GDNF (Sigma Aldrich, St. Louis, MO, USA), 100 U/mL human LIF (Millipore), 1% ES cell qualified FBS, and 1 µL/mL DL lactic acid (Sigma Aldrich, St. Louis, MO, USA) at 37 °C and 5% CO_2_ in air [3,15,16].

### 2.3. Fluidigm Dynamic Arrays for Gene Expression Analysis

The quantity of the vimentin gene expression (V2258) in the adult testicular cells was assayed by Fluidigm (dynamic array chips). Glyceraldehyde-3-phosphate dehydrogenase was utilized as a housekeeping gene for standardization. Cultured testicular cells were selected with a micromanipulator, and lysed with a lysis buffer solution containing 1.3 μL TE buffer, 9 μL RT-PreAmp Master Mix, 0.2 μL R.T./Taq Superscript III (Invitrogen, Waltham, MA, USA), 2.5 μL 0.2× assay pool, and 5.0 μL Cells Direct 2× Reaction Mix (Invitrogen, Waltham, MA, USA). The targeted transcripts were quantified using TaqMan real-time PCR on the Biomark real-time quantitative PCR system (Fluidigm-PCR). Cells were examined in two technical repeats. Ct evaluation was analyzed by GenEx software (version 5.4.2, MultiD Analyses, Gothenburg, Sweden) [3,16].

### 2.4. Immunocytochemical Staining

Testicular cells were fixed with 4% paraformaldehyde and then permeably with 0.1% Triton/PBS. The testicular cells were blocked with 1% BSA/PBS and incubated with primary antibody vimentin. Then, we used secondary antibodies specific for incubation fluorochrome species, and labeled cells were nuclear-counterstained treated with 0.2 μg/mL of 4′,6-diamidino-2-phenylindole (DAPI) dye. Labeled positive testicular cells were studied under confocal microscopy Zeiss LSM 700 (Oberkochen, Germany), and images were taken using a Zeiss LSM-TPMT camera (Oberkochen, Germany) [15,17].

### 2.5. Tissue Processing for Immunohistofluorescence Staining

Testicular cells of adult mice were washed with PBS and fixed in 4% paraformaldehyde. Dehydrated tissue samples were placed in Paraplast Plus and chopped with a microtome device at 10 μm thickness. Testis tissue sections were mounted on Superfrost Plus slides and kept at 25 °C until used. In the process of immunohistofluorescence (IMH) staining, samples were washed with xylene and gradually replaced with water in ethanol. Antigen retrieval was performed by heat-induced epitope retrieval at 94 °C for 18 min for tissue samples. The nonspecific binding site of tissue samples was blocked with 10% serum/0.3% Triton in PBS. As explained above, the experiment of immunofluorescence staining for these samples was continued [3].

### 2.6. Search Procedure and Data Preparation for Network Analysis

The spermatogenesis-related dataset was explored from a gene database (https://www.ncbi.nlm.nih.gov/gene/, accessed on 22 November 2021). The search procedure was spermatogenesis and “*Mus musculus*” (porgn: __txid10090). Then, the gene expression profile was collected in an excel file, and *p* ˂ 0.05 was considered for the selection of gene interactions and clusters.

### 2.7. Network Analysis of Protein–Protein Interactions (PPI)

The STRING (v.11) online tool was applied to predict protein–protein biological and functional interactions (https://stringdb.org/, accessed on 24 November 2021). Spermatogenesis genes with a significant role in vimentin were uploaded in the STRING online tool. The master regulators of vimentin and spermatogenesis-related signaling pathways were highlighted. Highlighted genes were imported into Cytoscape (version 3.8.2, Boston, MA, USA) for further analysis and protein–protein interaction network visualization.

### 2.8. Statistical Analysis

Trials were replicated at least three times. The average gene expression in all groups was calculated, and groups were evaluated with the Student’s *t*-test and compared with the nonparametric Wilcoxon–Mann–Whitney test. The difference among groups was considered to be statistically significant if *p* < 0.05.

### 2.9. Ethical Statement

In the current investigation, animal experiments were approved (Ir.ausmt.rec.1400.05) by the Amol University of Special Modern Technologies Ethics Committee.

## 3. Results

### 3.1. Vimentin Expression in Seminiferous Tubules by Immunohistochemistry

In this experiment, we first analyzed vimentin expression in the section of the mouse seminiferous tubules by immunohistochemistry (IMH). Analysis showed high expression of vimentin in differentiating germ cells in the adluminal and luminal compartments of seminiferous tubules, although it showed low expression of vimentin in undifferentiated cells located in the basal compartment of seminiferous tubules. In addition, immunohistochemical analysis detected vimentin expression near the basal membrane, showing that it was related to Sertoli cells (Figure 1). Subsequently, we used the SOX9 specific marker for the distinction of germ cells and Sertoli cells. IMH analysis detected high expression of SOX9 in the basal compartment that proved cytoplasm expression in Sertoli cells, and negative expression in differentiating germ cells (Figure 2). 

### 3.2. Testicular Cells Isolation and Vimentin’s Expression by Immunocytochemical Analysis

After enzyme digestion, isolated cells were cultured in the presence of growth factors. The characterization of isolated testicular cells was performed as described in our previous study [1]. We examined the expression of vimentin in two populations of differentiating germ cells (including primary and secondary spermatocytes, and round spermatids) and undifferentiated spermatogonia by immunocytochemical analysis (ICC). Images obtained from the confocal scanning UV laser microscope in ICC analysis demonstrated higher vimentin expression in differentiating germ cells than that in undifferentiated cells. 

In the following experiment, we utilized the DAZL specific marker for distinction between undifferentiated spermatogonia and differentiating germ cells (primary and secondary spermatogonia, and spermatids). DAZL was highly expressed in undifferentiated spermatogonia, although it was low in differentiating germ cells. Immunocytochemical analysis (ICC) showed that the vimentin is highly expressed in differentiating germ cells; it showed low expression in undifferentiated spermatogonia (Figure 3). The last experiment utilized the Ki67 specific marker to distinguish between differentiating cells (primary and secondary spermatogonia, and spermatids) and undifferentiated spermatogonia. Immunocytochemical analysis (ICC) showed high expression of Ki67 in differentiating germ cells, while we observed low expression in undifferentiated spermatogonia (Figure 4).

### 3.3. Analyses of Vimentin Gene Expression by Fluidigm Biomark System

In addition, Fluidigm real-time RT PCR results showed significant expression of vimentin mRNA in differentiating germ cells compared with that in undifferentiated spermatogonia (*p* < 0.05, Figure 5).

### 3.4. Protein–Protein Interaction Visualization of Vimentin in Spermatogenesis Process

The protein–protein interaction network was visualized with 945 gen using the STRING (v.11) database. It demonstrated that there was a close relationship between interaction and regulated vimentin in the spermatogenesis process. We observed a high level of interaction between Stat3, Mmp2, Trp53, Casp7, AURKB, Pik3r1, Ctnnb1, Lgals3, Cdkn1a, Snai1, and Pou5f1, and vimentin. In addition, there was a clear association among Trp53, Mmp2, Casp7, Stat3, and Pik3r1. Reactome and KEGG selected any spermatogenesis-related signaling pathway to highlight the master regulator of the spermatogenesis pathways. There was strong correlation between the highlighted genes as is shown in Figure 6.

## 4. Discussion

We showed a variable pattern in vimentin expression from self-renewal to differentiation processes. Immunohistochemical, immunocytochemical, and Fluidigm real-time PCR analyses showed that vimentin is expressed in different stages of spermatogenesis in vivo and in vitro. Our immunohistochemical and immunocytochemical analyses specified the localization of vimentin-positive cells in the center of testicular and vimentin-negative cells in the basal compartment of seminiferous tubules of adult testes. Protein analyses using immunohistochemical and immunocytochemical staining showed high expression of vimentin in differentiating germ cells, and low expression in undifferentiated spermatogonia. Previous experiments investigated limiting vimentin to decrease undifferentiated pluripotent embryonic stem cells (mESCs) [18] and spermatogonia stem cells [19,20]. Another study indicated that the absence of vimentin impairs the spontaneous differentiation of SSCs to the spermatogonia phenotype in seminiferous tubules [18,21]. A recent study demonstrated that mouse embryonic stem cells (mESC) from vimentin –/– mice reduce embryoid body processes. Mouse embryonic fibroblasts lacking vimentin are challenging to differentiate [18,22]. During the XII–XV stages of the epithelial cycle, Sertoli cells confirmed a response within the perinuclear location, and vimentin-positive extensions are a system toward developing spermatid bundles. During the VI–XI stages, these extensions were narrow and small [23]. Monoclonal antibodies to vimentin intermediate filament gave a granular reaction in the peripheral region of the flagellum [24,25,26,27]. Identifying numerous marker genes has enabled each developmental system to be identified at the cellular and molecular levels. This study performed immunofluorescence (IMF) analyses during the adult testicular process and in vitro to depict vimentin expression within spermatogenesis. Increasing cytoplasmic volume and cell organization are related to the expression of vimentin in cell increases (Figure 7). SSCs have low vimentin levels, which grow early in differentiation and later change according to tissue, and this process continues in the early stages of differentiating germ cells [25,28]. These results demonstrated a change in vimentin gene expression in spermatogonia during differentiation. Vimentin is thus upregulated during the differentiation stages. 

There was a close relation between the interaction and regulation of vimentin in the spermatogenesis process. There was high interaction among Vim, Stat3, Mmp2, Trp53, Casp7, AURKB, Pik3r1, Ctnnb1, Lgals3, Cdkn1a, Snai1, and Pou5f1. Trp53, AURKB, Cdk2, Ccnb2, and Bub1 are critical regulators of chromosomal segregation in meiosis [29,30]. Vimentin is the master regulator of spermatogenesis pathways in meiosis. Vimentin coregulation may thus increase the meiotic phase.

There was a close relationship between interaction and regulated vimentin in the spermatogenesis process. There was high interaction among Stat3, Mmp2, Trp53, Casp7, AURKB, Pik3r1, Ctnnb1, Lgals3, Cdkn1a, Snai1, and Pou5f1. In addition, there was a clear association between Trp53, Mmp2, Casp7, Stat3, and Pik3r1. Reactome and KEGG selected any spermatogenesis-related signaling pathway to highlight the master regulator of the spermatogenesis pathways. There was powerful correlation among the highlighted genes.

Due to the lack of information about vimentin expression of the different spermatogenic stages, we used datasets related to protein–protein interaction members in this study (Figure 7). Gene ontology analysis of gene expression demonstrated that different biological and functional pathways are involved in vimentin expression at a different stage of spermatogenesis. However, the increased expression of this protein is related to cell differentiation. Recently, vimentin expression has been related to SSC differentiation, localization, and cell surface receptor signaling pathways. In the study, biological and functional analysis showed increased vimentin expression following the Stat3, Mmp2, Trp53, Casp7, AURKB, Pik3r1, Ctnnb1, Lgals3, Cdkn1a, Snai1, and Pou5f1 genes (Table 1). The table shows gene and biological function in differentiating germ cells.

Many proteins are cofunctional with vimentin; for example, Stat3 regulates SSC differentiation by STAT3 signaling pathway [31]. TP53 expressions are related to the main phase regulating the meiotic system in the pachytene stage [3]. Some recent studies showed that AURKB and AURKC ensure the organization of chromosomal restructuring during the MI/G2 transition during the spermatogenesis process, and they maintain ordered system meiosis that is critical for accurate chromosome segregation [6]. Snai1 mediates developmental changes during cellular transitions in spermatogenesis [9]. Another protein cofunction with vimentin is Mmp2, which specifically regulates the survival or apoptosis of spermatocytes [5]. All of the following functions are related to the vimentin intermediate filament (Table 1).

These results suggest vimentin cofunction and cointeraction with SSC differentiation, cell surface, receptor signaling pathway, regulation of localization, organelle organization, and cell differentiation by the Stat3, Mmp2, Trp53, Casp7, AURKB, Pik3r1, Ctnnb1, Lgals3, Cdkn1a, Snai1, and Pou5f1 genes. Moreover, the up- or downregulation of those genes may increase differentiation spermatogenesis development (Figure 8). Further research can provide more detailed data in this area.

## 5. Conclusions

These results indicate that vimentin is an essential intermediate filament of testicular germ cells for differentiation, and it is upregulated in the differentiation stages of spermatogenesis. Analyses confirmed that vimentin is highly expressed in the center of seminiferous tubules and the differentiated section. It seems that this is highly expressed during differentiation, and its increase is associated with differentiation. In this study, we established a culture and identification system for mouse spermatogonial stem cells, and provided a reference for advanced studies on other mammalian and even human SSCs.

## Figures and Tables

**Figure 1 animals-12-00268-f001:**
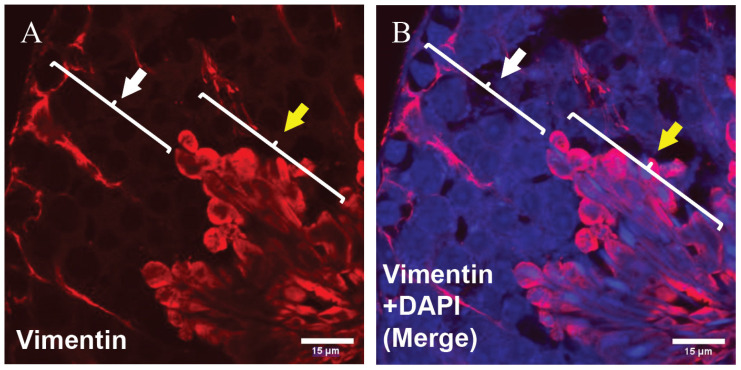
Immunohistochemical characterization of vimentin intermediate filament in seminiferous tubules of adult mice. (**A**) High expression of vimentin in differentiating germ cells located in the middle compartment of seminiferous tubules (yellow arrow), although it showed low expression in undifferentiated cells located in the basal compartment of seminiferous tubules (white arrow). (**B**) Merged image with DAPI. Vimentin, red and DAPI, blue (scale bar: 15 μm).

**Figure 2 animals-12-00268-f002:**
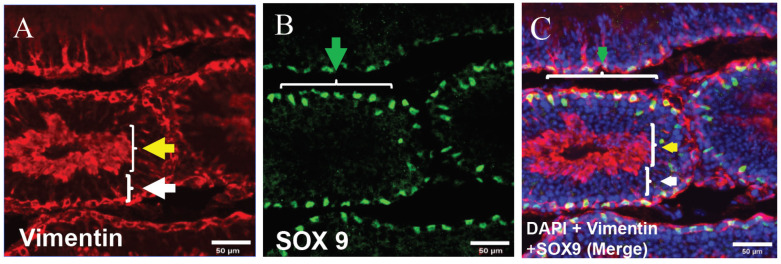
Vimentin and SOX9 positive cell localization in seminiferous tubules of mice by immunohistochemical (IMH) analysis. (**A**) Vimentin expression in basal region was low (white arrow), and high in the luminal compartment (yellow arrow). (**B**) SOX9 expression in Sertoli cells (green arrow). (**C**) Merged image with DAPI. Vimentin, red; SOX9, green; and DAPI, blue (scale bar: 50 μm).

**Figure 3 animals-12-00268-f003:**
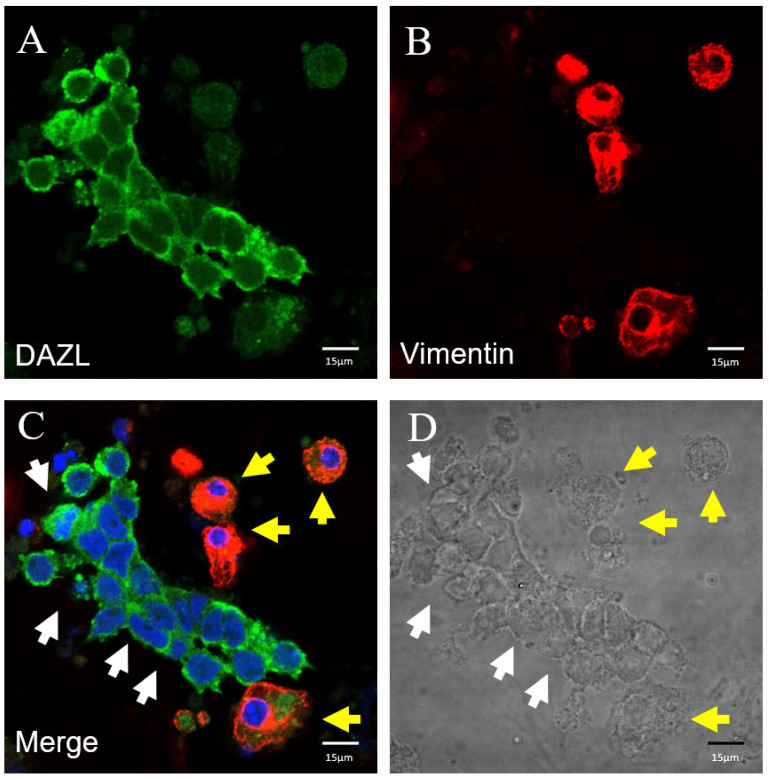
Immunocytochemical analysis of vimentin and DAZL. (**A**) Green fluorescence shows DAZL expression; (**B**) red fluorescence shows vimentin expression; (**C**) merged image with DAPI. Undifferentiated spermatogonia (white arrow) and differentiating germ cells (yellow arrow). Vimentin, red; DAZL, green; and DAPI, blue (scale bar: 15 μm). (**D**) Bright field. (scale bar: 15 μm).

**Figure 4 animals-12-00268-f004:**
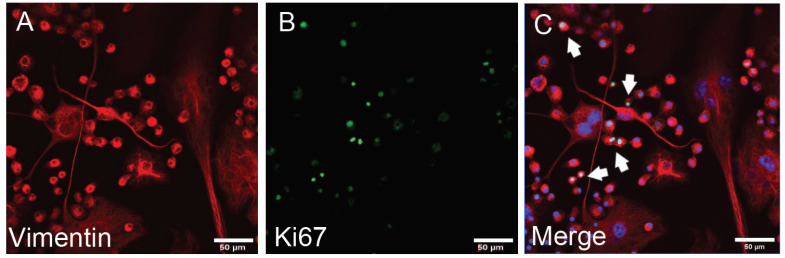
Immunocytochemical analysis of vimentin and Ki67. (**A**) Green fluorescence shows Ki67 expression; (**B**) red fluorescence shows vimentin expression; (**C**) merged image with DAPI (differentiating cells shown with a white arrow). Vimentin, red; Ki67, green; and DAPI, blue (scale bar: 50 μm).

**Figure 5 animals-12-00268-f005:**
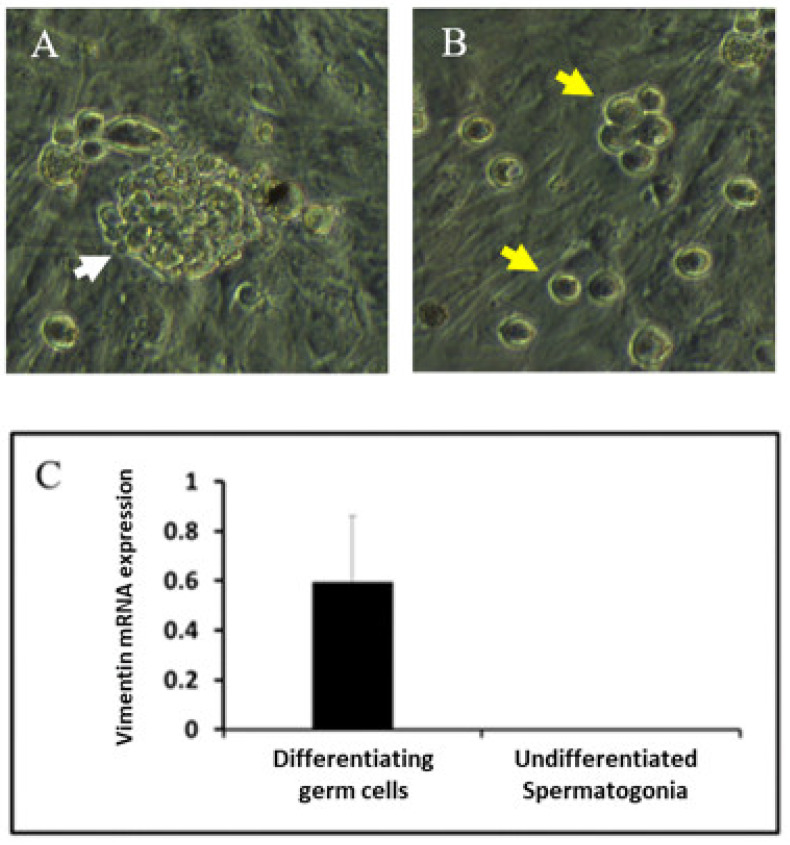
Bright-field image of undifferentiated spermatogonia and differentiating germ cells, and comparison of mRNA levels between them. (**A**) Bright-field image of undifferentiated spermatogonia (white arrow). (**B**) Bright-field image of differentiating germ cells (yellow arrow). (**C**) Fluidigm quantitative polymerase chain reaction analysis for vimentin expression in two populations of testicular cells (undifferentiated spermatogonia and differentiating germ cells). High mRNA expressed in differentiating germ cells, and no expression of mRNA detected in undifferentiation spermatogonia. At least *p* < 0.05 versus other groups. Data presented as mean ± SD (scale bar: 15 μm).

**Figure 6 animals-12-00268-f006:**
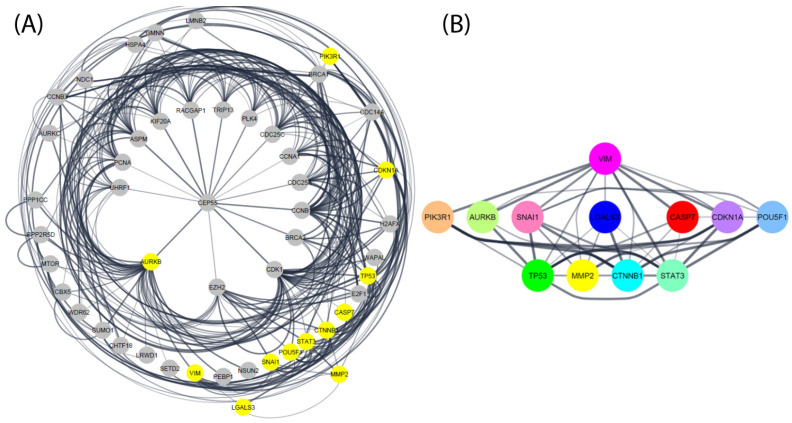
STRING protein–protein interaction network based on reactome and KEGG pathways. (**A**) Total genes involved in vimentin in spermatogenesis process. (**B**) Direct linkage of genes involved in vimentin.

**Figure 7 animals-12-00268-f007:**
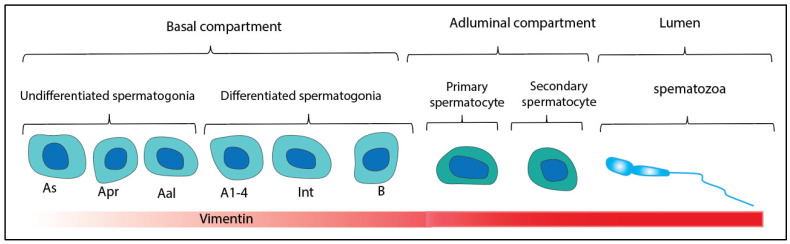
Scheme of vimentin intermediate filament gene expression in mouse spermatogenesis system. Vimentin expression is low in undifferentiated spermatogonia and increases in differentiating germ cells. It increases more in developmental stages such as primary and secondary spermatocytes, spermatids, and spermatozoa.

**Figure 8 animals-12-00268-f008:**
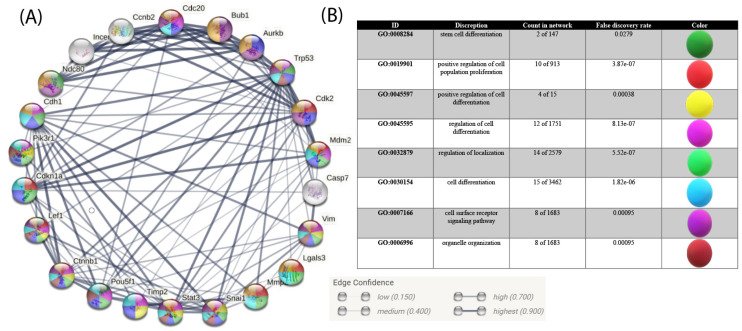
STRING protein–protein interaction network based on reactome and KEGG pathways, and STRING database. (**A**) Protein–protein interaction of spermatogenesis-regulated demonstrated a meaningful coexpression. Highlighting nodes by cell differentiation, cell surface receptor signaling pathway, and other functions demonstrating hub genes. (**B**) Chosen regulation function interaction with vimentin.

**Table 1 animals-12-00268-t001:** Close relationship protein–protein interaction between differentiation and regulation in spermatogenesis process.

Genes	Discription	Ref
Vim	Vimentin; intermediate filament found in spermatogenesis cells. Vimentin is connected to the mitochondria and endoplasmic reticulum either laterally or terminally	[2]
Stat3	Signal transducer and activator of transcription 3; regulation of mouse SSC differentiation by STAT3 signaling.	[3,4]
Mmp2	matrix metalloproteinase 2, 72 kDa type IV collagenase; CD147 regulates migration of SSC and spermatocytes via induction of MMP2 production; it specifically regulates survival/apoptosis of spermatocytes.	[5]
Trp53	Tumor antigen p53; acts as a tumor suppressor in many tumor types; p53 gene (TP53) is essential in regulating apoptosis during spermatogenesis. TP53 expressions are associated with the main phase regulating meiotic progression with a peak in the pachytene stage.	[6]
Casp7	Caspase-7; caspase 7 has a nonapoptotic function that participates in differentiating germ cells and helps in apoptotic differentiating germ cellsdysfunction.	[7]
AURKB	Aurora Kinase B; some mitosis studies showed that AURKB plays an essential role in destabilizing kinetochore-microtubule attachments at sister kinetochores required for coordinating SC disassembly, chromosome compaction, and chromosome segregation during spermatogenesis. AURKB and AURKC ensure the organization of chromosome restructuring events during the MI/G2 transition during the spermatogenic process, and they are vital for accurate chromosome segregation.	[8,9]
Pik3r1	Phosphatidylinositol 3-kinase regulatory subunit alpha; binds to activated (phosphorylated) protein-Tyr kinases through its SH2 domain and acts as an adapter, mediating the association of the p110 catalytic unit to the plasma membrane.	[10]
Ctnnb1	Catenin beta-1; WNT binds CTNNB1 pathway mediates proliferation of progenitor cells and spermatogonial stem cells in spermatogenesis process.	[11]
Lgals3	Galectin-3; Galactose-specific lectin which binds IgE. May mediate with alpha-3, beta-1 integrin the stimulation by CSPG4 of endothelial cell migration.	[12]
Cdkn1a	Cyclin-dependent kinase inhibitor 1; involved in p53/TP53 mediated inhibition of molecular proliferation process in response to DNA damage. The role of Cdkn1a in SSC differentiation and renewal. Cdkn1a is expressed in primordial germ cells and SSC, with a direct effect of cell cycle arrest and p53 expression on levels of Cdkn1. It is expressed in spermatocytes and spermatids.	[13,14]
Snai1	Zinc finger protein SNAI1; SNAI1 is seen in nuclei of spermatocytes, round spermatids, and elongated spermatids. Snail factors are critical for transcriptional regulation, cellular migration, signal transduction, and chromatin remodeling in the spermatogenesis process.	[15]
Pou5f1	POU domain, class 5, transcription factor 1; POU5F1 regulates pluripotency during normal development. pou5f1/POU5F1 functions as a critical role in differentiation by regulating cells that can develop pluripotent potential. The study shows that POU5F1 downregulation in differentiating spermatogonia is an essential step for the spermatogenesis process.	[16]
Cdh1	Cadherin-1; cadherins are calcium-dependent cell adhesion proteins.	[17]
Cdc20	Cell division cycle protein 20 homolog initiates sister-chromatid separation and activating subunit of the anaphase-promoting complex/cyclosome (APC/C) by ordering the destruction of two key anaphase inhibitors, cyclin B1 and securin, at the transition from metaphase to anaphase.	[18]
Ndc80	Kinetochore protein NDC80; functions as an essential kinetochore-associated NDC80 complex.	[19]
Incenp	Inner centromere protein is a component of the chromosomal complex (CC), a complex that functions as a vital regulator of mitosis. Incenp is required in mitosis for chromosome condensation and spindle attachment.	[20]
Cdk2	Cyclin-dependent kinase 2; serine/threonine-protein kinase involved in the control of the cell cycle. Cdk2 was highly expressed in spermatocyte process.	[21]
Bub1	Bub1 is a key chromosomal segregation regulator in meiosis and mitotic checkpoint serine/threonine-protein kinase.	[22]
Ccnb2	G2/mitotic-specific cyclin-B2 is the master control of the cell cycle at the mitosis transition in the spermatogenesis process.	[23]

## Data Availability

The data presented in this study are available on request from the corresponding author.

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
