# Peer review of "Testicular Localization and Potential Function of Vimentin Positive Cells during Spermatogonial Differentiation Stages"

_animals, 2022, doi:10.3390/ani12030268_

Round 1

Reviewer 1 Report

This is a good study, and it is well presented. I only perceive some repetitions in the text, and that there are still more studies to have more conclusive results. But it advances in knowledge.

Author Response

Manuscript title: Testicular localization and potential function of vimentin positive cells during spermatogonial differentiation stages

Manuscript ID: animals-1454025

We should express our gratitude for the reviewer comments and have improved and corrected our manuscript accordingly.

Reviewer 1: This is a good study, and it is well presented. I only perceive some repetitions in the text, and that there are still more studies to have more conclusive results. But it advances in knowledge. We made the corrections.

Reviewer 2 Report

Dear Authors

Some parts of the manuscript could benefit from a minor language revision, while the structure (or lack of) in some sentences makes some parts possible very difficult to understand. Thus, the whole text (including the abstract) requires an extensive language revision (perhaps by a native speaker).

As the whole manuscript requires a thorough language revision, I will leave the following five comments as some examples:

  • Line 44: Sentence is hard to understand; should the word “differentiate SSC” be “the differentiated SSCs”?

Line 74: “In this experimental study, protocols for animal care and surgical intervention of 74 adult mice in this study were approved by Amol University of Special Modern Technologies, Amol, Iran.”

  • as another example of the required language revision: “In this experimental study” and “in this study” are the same, and one needs to be excluded.

Line 77: “Mouse testis was collected…”

  • was it only one testis from each mouse, or both?  If both, the term “testis” (singular) should be replaced with “testes” plural. This should be revised throughout the text accordingly.

Line 80: “Testicular cells … for 10 minutes.”

  • Should be changed to: “Testicular cells were collected by enzymatic digestion in a solution containing Dispase (0.5 mg/ml) (Sigma Aldrich, USA), collagenase IV (0.5 mg/ml) (Sigma Aldrich, USA) and DNase (0.5mg/ml) (Sigma Aldrich, USA) in HBSS buffer with Ca2+ and Mg2+ (PAA, USA) at 37°C for 10 minutes.”

Line 83: “then washed with…”

  • The sentence needs to be restructured as a sentence on its own or connected to the previous sentence. It is not grammatically correct and difficult to follow.

Line 40: please change “paired (Apr)” to “A paired (Apr)” to show the same abbreviation structure as “A single (As)” and “A aligned (Aal)”.

Some general comments to be considered on the methods and results:

Line 140-144:

The statistical analysis process is a little unclear.  It has been reported that “the groups were evaluated utilizing one-way analysis of variance (ANOVA), continued with the Student's t-test”, which are parametric tests, and then continued to say “and compared with the non-para-143 metric Wilcoxon-Mann-Whitney test.”

  • Were the data normally or not normally distributed?

Line 175: “… in our previous study.”

  • Please add a reference to the “previous study”.

The whole discussion section should be reviewed after language revision.

Line 322:

Author contributions need to be revised. There are some elements from the “template” which have not been changed or deleted.

Author Response

Manuscript title: Testicular localization and potential function of vimentin positive cells during spermatogonial differentiation stages

Manuscript ID: animals-1454025

We should express our gratitude for the reviewer comments and have improved and corrected our manuscript accordingly.

Reviewer 2: Some parts of the manuscript could benefit from a minor language revision, while the structure (or lack of) in some sentences makes some parts possible very difficult to understand. Thus, the whole text (including the abstract) requires an extensive language revision (perhaps by a native speaker).

As the whole manuscript requires a thorough language revision, I will leave the following five comments as some examples:

Line 44: Sentence is hard to understand; should the word “differentiate SSC” be “the differentiated SSCs”? Done.

Line 74: “In this experimental study, protocols for animal care and surgical intervention of 74 adult mice in this study were approved by Amol University of Special Modern Technologies, Amol, Iran.” as another example of the required language revision: “In this experimental study” and “in this study” are the same, and one needs to be excluded. Done.

Line 77: “Mouse testis was collected…” was it only one testis from each mouse, or both?  If both, the term “testis” (singular) should be replaced with “testes” plural. This should be revised throughout the text accordingly. Done.

Line 80: “Testicular cells … for 10 minutes.” Should be changed to: “Testicular cells were collected by enzymatic digestion in a solution containing Dispase (0.5 mg/ml) (Sigma Aldrich, USA), collagenase IV (0.5 mg/ml) (Sigma Aldrich, USA) and DNase (0.5mg/ml) (Sigma Aldrich, USA) in HBSS buffer with Ca2+ and Mg2+ (PAA, USA) at 37°C for 10 minutes.” Done.

Line 83: “then washed with…” The sentence needs to be restructured as a sentence on its own or connected to the previous sentence. It is not grammatically correct and difficult to follow. Done.

Line 40: please change “paired (Apr)” to “A paired (Apr)” to show the same abbreviation structure as “A single (As)” and “A aligned (Aal)”. Done.

Some general comments to be considered on the methods and results:

Line 140-144:

The statistical analysis process is a little unclear.  It has been reported that “the groups were evaluated utilizing one-way analysis of variance (ANOVA), continued with the Student's t-test”, which are parametric tests, and then continued to say “and compared with the non-para-143 metric Wilcoxon-Mann-Whitney test.” Were the data normally or not normally distributed? Done; Our comparison was with normal data, and the results were standard.

Line 175: “… in our previous study.” Please add a reference to the “previous study”. The whole discussion section should be reviewed after language revision. Done.

Line 322:Author contributions need to be revised. There are some elements from the “template” which have not been changed or deleted. Done.

Reviewer 3 Report

In this manuscript, the authors studied the expression and localization of vimentin in mouse testis. Although the results are interesting, the whole manuscript should be carefully revised. My suggestions and comments are listed below, in no particular order:

Title:  it is hard to understand what the authors tried to express. Do the authors mean" localization and potential function of vimentin positive cells in mouse testis”?

What is the “differentiation stages”? Do the authors mean spermatogonial differentiation?

SSCs are supported by Sertoli cells, peritubular myoid cells and other somatic cells.

Differentiate SSC divides into Primary spermatocytes… a very confusing sentence.

Vimentin is critical to differentiate SSCs. Again, this sentence should be revised.

SSCs are rare cells reside within the undifferentiated spermatogonial population. To call the isolated cells “SSCs” is simply not accurate. Most likely, primary cultures of undifferentiated spermatogonia.

Figure 4, how did the authors identify undifferentiated spermatogonia and differentiated spermatogonia? Without proper makers, for example, Lin28, PLZF for undiff. Spg, Kit, Stra8 for diff. spg. It is impossible to tell.

Figure 5. Some of the Vimentin positive cells appear to be round spermatids. Please confirm the identity of these cells.

Figure 3. How the differentiating spermatogonia was identified?  Spermatogonia?

Figure 6 is not supported by the data. Is vimentin expressed in spermatocytes? It appears that vimentin is in Sertoli cells and spermatids.

Author Response

Manuscript title: Testicular localization and potential function of vimentin positive cells during spermatogonial differentiation stages

Manuscript ID: animals-1454025

We should express our gratitude for the reviewer comments and have improved and corrected our manuscript accordingly.

Reviewer 3: In this manuscript, the authors studied the expression and localization of vimentin in mouse testis. Although the results are interesting, the whole manuscript should be carefully revised. My suggestions and comments are listed below, in no particular order:

Title:  it is hard to understand what the authors tried to express. Do the authors mean" localization and potential function of vimentin positive cells in mouse testis”? Done.

What is the “differentiation stages”? Do the authors mean spermatogonial differentiation? Done.

SSCs are supported by Sertoli cells, peritubular myoid cells and other somatic cells. Done.

Differentiate SSC divides into Primary spermatocytes… a very confusing sentence. Done.

Vimentin is critical to differentiate SSCs. Again, this sentence should be revised. Done.

SSCs are rare cells reside within the undifferentiated spermatogonial population. To call the isolated cells “SSCs” is simply not accurate. Most likely, primary cultures of undifferentiated spermatogonia. Done.

Figure 4, how did the authors identify undifferentiated spermatogonia and differentiated spermatogonia? Without proper makers, for example, Lin28, PLZF for undiff. Spg, Kit, Stra8 for diff. spg. It is impossible to tell. According to our previous studies, we characterized undifferentiated spermatogonia and differentiated spermatogonia with specific markers that were not our main goal in the present study; So, we just used our last experiences using specific markers to identify undifferentiated spermatogonia from differentiated.

Azizi, H., Niazi Tabar, A. and Skutella, T., 2021. Successful transplantation of spermatogonial stem cells into the seminiferous tubules of busulfan-treated mice. Reproductive Health, 18(1), pp.1-9.

Azizi, H., Tabar, A.N., Skutella, T. and Govahi, M., 2020. In Vitro and In Vivo Determinations of The Anti-GDNF Family Receptor Alpha 1 Antibody in Mice by Immunochemistry and RT-PCR. International Journal of Fertility & Sterility, 14(3), p.228.

Figure 5. Some of the Vimentin positive cells appear to be round spermatids. Please confirm the identity of these cells. As our observations, we suggested that Vimentin expressed within spermatogonial differentiation stages; So, we explained that there is the possibility of expression of vimentin in all of the differentiated cells or just in some of them; But, it is important to mention that the exact explanation of cells that expressed Vimentin seem to be impossible. So, we can not say exactly which one of the differentiated cells (round spermatids or other cells) expressed Vimentin.

Figure 3. How the differentiating spermatogonia was identified?  Spermatogonia? In our immunocytochemistry analysis, we used DAZL to identify undifferentiated spermatogonia and then the expression of vimentin detected in the cells that didn’t express DAZL (mean differentiated). On the other hand, we have studied the characterization of undifferentiated spermatogonia and differentiated spermatogonia with specific markers.

Azizi, H., Niazi Tabar, A. and Skutella, T., 2021. Successful transplantation of spermatogonial stem cells into the seminiferous tubules of busulfan-treated mice. Reproductive Health, 18(1), pp.1-9.

Figure 6 is not supported by the data. Is vimentin expressed in spermatocytes? It appears that vimentin is in Sertoli cells and spermatids. As our main goal of the present study, we investigated the expression of Vimentin in germ cells during differentiation stages, but we did not investigate or explain the expression of Vimentin in somatic cells, and we did not assign any specifics for expression of Vimentin only in germ cells as evidence, we used Sox9 as Sertoli specific marker to make a distinction between somatic cells (Sertoli cells) and differentiated germ cells (that were our main goal of study).

Round 2

Reviewer 2 Report

Dear Authors

The manuscript has been significantly improved following the revision.

I have no further comments on the content, although there are still some minor grammatical errors that require some attention.

Author Response

Manuscript title: Testicular localization and potential function of vimentin positive cells during spermatogonial differentiation stages

Manuscript ID: animals-1454025

We should express our gratitude for the reviewer comments and have improved and corrected our manuscript accordingly.

Reviewer 2: The manuscript has been significantly improved following the revision. I have no further comments on the content, although there are still some minor grammatical errors that require some attention. We made the corrections.

Reviewer 3 Report

After reviewing the revised version of this manuscript, unfortunately, my major concerns have not been addressed. Several big flaws still exist. For example, how the authors tell the difference between the undifferentiated spermatogonial population and differentiating spermatogonial population? Therefore, I recommend rejection and hope the authors take time to carefully conduct additional experiments to provide solid evidence that this gene is involved in spermatogonial differentiation. 

Author Response

Manuscript title: Testicular localization and potential function of vimentin positive cells during spermatogonial differentiation stages

Manuscript ID: animals-1454025

We should express our gratitude for the reviewer comments and have improved and corrected our manuscript accordingly.

Reviewer 3:

After reviewing the revised version of this manuscript, unfortunately, my major concerns have not been addressed. Several big flaws still exist. For example, how the authors tell the difference between the undifferentiated spermatogonial population and differentiating spermatogonial population? Therefore, I recommend rejection and hope the authors take time to carefully conduct additional experiments to provide solid evidence that this gene is involved in spermatogonial differentiation.

Our previous studies characterized undifferentiated spermatogonia and differentiated spermatogonia with morphology-based selection, Fluidigm real-time RT-PCR, Immunocytochemistry, FACS, and Electron microscopic analysis. Also, our recent studies demonstrated them, and we summarized below:

1- We identified a simple and highly reproducible protocol for the isolation of SSCs with morphology-based selection by microscopy analysis and successfully established two types of undifferentiated and differentiating cells about 10 days after enzymatic digestion in cell extraction plate.

2- We picked up both two types (undifferentiated and differentiated) with micromanipulator.

3- Then, we expanded both cell populations on MEF and STO feeder layer. Two distinct types of differentiating and undifferentiated cells are obviously shown.

A

4- Electron microscopic analysis showed undifferentiated cells characterized by big nucleolus and small cytoplasm rim (high nucleus/cytoplasm ratio) (A). High nucleolus/cytoplasm ratio is an essential property of undifferentiated SSCs in testis. In contrast we observed that nucleus/cytoplasm ratio in differentiating cells was low and they were packed with vesicles and other cell organelles (B).

B

5- Molecular analysis indicated two populations; Fluidigm real-time RT-PCR, FACS, and Immunocytochemistry results showed that undifferentiated SSCs clearly express germ cells markers while differentiating cells only partially express the typical germ cell profile of SSCs.

Furthermore, Immunohistochemistery (IMH) and immunocytochemistery (ICC) analyses of zbtb16 indicate high expression of the zbtb16 in cells located on the basement membrane (undifferentiated) of the seminiferous tubules in culture (A2, B2) while differentiating cells shown low expression. Merging blue DAPI and zbtb16 (A3, B3) (Scale bar = 50 μm).
